# Changes in reproductive behavior associated with the perception and individual experiences of the COVID-19 pandemic

**Jitka Slabá** [ID]*

Department of Demography and Geodemography, Prague, Czechia

* slabaji@natur.cuni.cz

**Data Availability Statement:** Data are available on GGP Data Archive. The free registration is needed. https://doi.org/10.57865/K867-AH67.

**Funding:** This output was supported by the NPO "Systemic Risk Institute" number LX22NPO5101,

## Abstract

This study evaluates the impacts of the COVID-19 pandemic on the reproductive behavior of men and women during the most restrictive period of the pandemic in Czechia. At the end of this period, data was collected for the Czech GGS COVID Pilot–Follow-up Study (April 2021), which included additional questions on reproductive plans and perceptions of the pandemic related to fertility. The study focuses in detail on the evaluation of the favorability of fertility during this period by considering the impact of the pandemic on the lives of individuals in various life areas. It subsequently attempts to determine to what extent this evaluation and personal experiences of the pandemic affected reproductive behavior (the intended number of children, current attempts to conceive and short-term fertility intentions). In summary, in most cases the respondents considered the most severe period of the pandemic to be unfavorable in terms of childbirth. Women provided an overall negative assessment of the favorability of childbirth in this period, which was reflected in a reduction in the planned number of children, while the men who considered this period favorable declared a higher chance of short-term fertility intentions.

## Introduction

The COVID-19 pandemic, which has exerted a serious global impact since the beginning of 2020, has been reflected in a significant increase in mortality levels [1], and its impact on fertility has been discussed by experts since its outset [2, 3]. The beginning of the pandemic was characterized by a general fear of the unknown and uncertainty as to its development, which resulted in people limiting their interaction in society; face-to-face meetings all but ceased as did the provision of non-essential services. In addition to concerns about health and engaging in society, uncertainty about the future functioning of society deepened, including with respect to financial security. The sociological literature refers to the pandemic period as a time of multiple crises [4], which had the potential to exacerbate existing and/or create new patterns of social inequality and, consequently, to influence reproductive behavior.

It is not unreasonable to assume that the pandemic affected reproductive behavior both negatively and positively, one of the former comprising a growing perception of uncertainty both at the general societal level in terms of "what will happen now to our society?" to the

funded by European Union - Next Generation EU (Ministry of Education, Youth and Sports, NPO: EXCELES) and the Charles University UK UNCE/ HUM/018 programs. The funders had no role in study design, data collection and analysis, decision to publish, or preparation of the manuscript.

**Competing interests:** The author has declared that no competing interests exist.

individual level of concerns about stability or even the complete loss of income and thus life security. In the context of reproductive behavior, limitations in terms of access to and the comfort level of health care and contact with friends and relatives during the pandemic presented potentially unfavorable factors. Conversely, limitations in terms of daily life responsibilities due to the constraints placed by the pandemic allowed for a deepening of relationships between partners and the increased involvement of both parents (more often this advantage concerned fathers) in caregiving, which may have led some to feel that the effects of the pandemic were in fact favorable for parenting [5].

The impact of the anti-pandemic measures on behavior was, on the whole, greater than that of the number of cases and deaths. In retrospect, and perhaps partly due to the measures adopted, the impact of the pandemic at the beginning of 2020 in Czechia was relatively mild. Thus, the consequences of the disease in the spring of 2020 were marginal in terms of the overall view of the development of the pandemic. The anti-pandemic measures were relaxed significantly in Czechia during the summer of 2020, and their reintroduction at the beginning of autumn was met with significant public opposition. The COVID-19 situation deteriorated once more during the autumn and winter months, finally culminating in the following spring. The period 27 February to 11 April 2021 witnessed the imposition of the strictest measures to date in Czechia aimed at combatting the spread of the pandemic [6]. At the end of this period, at which time the Czech population had had more than a year of experience with COVID-19, a follow-up pilot survey was conducted as part of the Generation and Gender Programme project [7, 8] in which, in addition to the second round of the core questionnaire, new questions were included that allowed for the study of the reproductive behavior of the population in the context of the pandemic. Thus, the timing of the collection of the data provided an opportunity to examine the impact of the most restrictive pandemic measures on the reproductive behavior of Czech men and women.

This study contributes to the understanding of the impact of the restrictive measures and the overall perception and experiences of the pandemic on reproductive behavior. Using the Czech GGS COVID Pilot—Follow-up Study dataset [9], the study poses the following questions: 1) How did people's experiences of the COVID-19 pandemic affect their assessment of the pandemic period in terms of fertility, and 2) How did their experience of the COVID-19 pandemic and their assessment of the pandemic period in terms of fertility affect their reproductive behavior?

## Theoretical background

Changes in the reproductive behavior of high-income countries are usually discussed according to two main theoretical concepts, the most well-known of which is the Second demographic transition [10–12], which relates fertility development to changes in societal values and norms such as the shifting position of women, more reliable and available contraception and selecting a life style according to one's own priorities and opportunities. However, the anti-pandemic measures limited these options to a significant extent, and it is not unreasonable to assume that for part of the population, starting (or expanding) a family moved up many people's list of priorities. In addition to the change in priorities [13], the enhancement of the reconciliation of work and family responsibilities may have contributed positively to fertility attitudes, for example via the option to work from home [13, 14] and the forced reduction of working hours, which comprised an official supportive measure in many European countries during the pandemic [13, 15]. Voicu and Badoi [16] suggest that this reconciliation may have increased the fertility of women who were forced to stay at home to care for their children due to the pandemic, and thus were forced to take on the position of caregiver due to the

reduced availability of formal and informal childcare services [5]. It should be noted here that although the extra demands for childcare impacted both mothers and fathers, most of the extra responsibilities were shouldered by women [17–20]. On the other hand, the extra burden of caring for children, together with the limited opportunities for self-realization, as well as changes in the life situations of individuals in terms both of the quality of the relationship [21, 22] and employment and income stability [4] may have resulted not only in a preference for parenthood but also, in the case of the deterioration of conditions, may have acted to reduce or delay reproduction.

An alternative concept concerning fertility changes in high-income countries considers changes to the previously straightforward transition from childhood to adulthood. The prolonging of education and the need to stabilize one's position in the labor market has led to increased uncertainty during early adulthood, thus resulting in the postponement of fertility to later ages [23, 24]. The pandemic disrupted existing certainties, and it is possible that women and men decided to postpone childbearing to more stable times or reduced or even abandoned their fertility plans altogether. Recent studies have pointed out that, in addition to the influence of past experience and current perceptions on fertility behavior, future expectations also play a role (Narrative framework [25]).

Studies conducted before the pandemic determined that the lower financial stability of individuals is associated with the lower probability of having or realizing fertility intentions [26–29]. It was indeed confirmed during the pandemic that a feeling of unstable financial security is associated with a greater chance of delaying fertility [30], while occupying more vulnerable job positions increases the chance of abandoning one's reproductive plans altogether [31]. In addition to the financial stability of the individual and the household, the employment sector in which the individual works also plays an important role since it is assumed that changes in reproductive intentions due to the pandemic varied according to occupation [32], i.e. the extent to which the individual's labor sector was affected by the pandemic, whether remote working was possible, etc.

Finally, an Italian study [31] pointed out that the effects of the pandemic varied according to its impact, with more affected regions witnessing more intense changes in reproductive intentions. Thus, following this logic, the period March and the first half of April 2021 was selected for the study of the potential effects of the pandemic on reproductive behavior in Czechia, since this period mirrored the mindset of the population in the most restricted period of the pandemic.

## The study

This study focuses on the reproductive behavior of Czech men and women during the pandemic; in addition to the respondents' overall experience of the pandemic, we took into account their assessment of whether they considered April 2021 to be favorable or unfavorable in terms of reproductive plans. Subsequently, we considered to what extent the individual's experience and assessment of favorability was related to their reproductive behavior concerning three areas: a reduction in the planned number of children, current efforts to conceive a child, and short-term fertility intentions.

## Data and methods

The empirical part of the study is based on the Czech GGS COVID Pilot—Follow-up Study (part of the second round of the Generation and Gender Programme (GGP) [9]. The first wave, the Czech GGS pilot study [33] was conducted in the period 9 December 2020 to 9 February 2021. The data was collected via the quota sampling method with controls for the sex

and age of the respondents. Only men and women between the ages of 18 and 69 were interviewed (via computer-assisted web interviews). A total of 1313 respondents completed the Czech GGP questionnaire during the first wave. The same respondents were invited to participate in the second wave from 4 to 19 April 2021. The second questionnaire repeated 23 questions from the first wave and posed 17 additional questions that focused primarily on the impact of COVID-19 on the Czech population. 1187 respondents participated in the second wave of the pilot study. 782 respondents aged 18 to 49 were considered in the analysis.

This study considers the five questions on fertility and the impact of COVID-19 on fertility plans. Information on the age of the respondents at the time of the collection of the data and the current number of children were used as the control variables. The inclusion of age in the analysis as one of the control variables comprised two variants, the age and the age-square, aimed at reflecting the fact that the association of age and reproductive behavior is not linear. The current number of children included both biological and adopted children and was recoded to the binary variables 1 = none or one child and 0 = two children and more; the latter was considered as the reference value. The two-children family model remains the most popular option [34, 35] and the expectation of additional fertility intentions are lower for respondents with two and more children than for those who are childless or who have only one child [36]. The analysis was conducted independently for men and women since it was anticipated that the differences in the life areas between the two would impact their reproductive behavior. This paper use five questions from the Follow-up questionnaire, they are as follows (in the order in which they appeared in the questionnaire):

1. How has the COVID-19 pandemic affected the following areas of your life? (Your financial situation; Financial situation in your household; Job security; Your working conditions, including working arrangements; Your state of health; Quality of your partner relationship; Relationship with other family members; Provision of domestic duties). The options were: significantly worsened; rather worsened, no change, rather improved; significantly improved; not applicable (recoded to no change for analysis purposes). The battery of questions was reduced to two variables employing principal component analysis. The statement "Your state of health" was excluded during the reduction process since its communalities value was lower than 0.4. The extracted variables explained 61.3% of the overall variability. The first component explained 42.4% of the overall variability, and it corresponded mainly to the first four statements and was labelled 'Job & Finances'. The second component explained 18.9% of the overall variability, and it corresponded mainly to the last three statements, and was labelled 'Family life'. These values were entered into the binary logistic models as continuous variables. The factor scores of each component had a higher value if the questioned area of life was declared as having improved the overall situation.

2. Regardless of your plans, do you rate the considered period of the pandemic (April 2021) as favorable or unfavorable for childbirth? The options were: very favorable; rather favorable; neither favorable nor unfavorable; rather unfavorable; very unfavorable. This question was entered into the analysis as a dependent and as an independent variable. As a dependent variable, it was entered as a binary variable, 1 = very favorable and rather favorable, 0 = other options. As an independent variable, it was entered as a continuous variable, where 1 = definitely unfavorable and 5 = definitely favorable.

3. Are you or your current partner attempting to become pregnant? The options were: yes (= 1); no (= 0).

4. Do you intend to have a/another child during the next three years? Please take into account only biological children. The possible options were: definitely not; probably not; unsure;

probably yes; definitely yes; currently expecting a child. This question was entered as a binary variable: 0 = the definitely not, probably not and unsure options, and 1 = definitely and probably yes. The respondents who chose the option "currently expect a child" were excluded from the analysis.

5. Have your plans for the number of children changed in the last three months due to the COVID-19 pandemic? The options were: Yes, I now wish to have fewer children (including the desire to remain childless); Yes, I now wish to have more children; No, my wishes regarding the number of children planned have not changed. This question was entered as a binary variable: 0 = no change or intend to have more children, 1 = intend to have fewer children. As indicated by the recoding, the object of interest concerned the reduction of fertility intentions due to the pandemic.

SPSS software was used for the analysis.

## Results

### Individual experiences of the pandemic and the favorability of the considered period for childbirth

Based on evaluation of the impact of the first year of pandemic on selected areas of life, the situations have been slightly worsened. When the evaluation of each statement was compared independently, the average was between 2.59 and 2.92 (2 = the situation rather worsened; 3 = no change in the situation). The Job & finances-related statements indicated a slightly worse situation for the women respondents then for men. However, the only significant difference (within the 95% confidence interval) concerned the evaluation of "job security", where the average evaluation was 2.59 for women and 2.76 for men. The differences between the sexes in terms of the evaluation of the Family life-related statements reveal slightly worse impact of the pandemic on men then on women. Nevertheless, the differences were insignificant.

Men rated the favorability of the considered period of the pandemic for childbirth more optimistically than women. 11% of the male respondents rated the situation as definitely unfavorable, 43% as rather unfavorable, 30% as neither unfavorable nor favorable, 13% as rather favorable, and 3% as definitely favorable. On the other hand, the female respondents rated the situation as follows: 18% as definitely unfavorable, 42% as rather unfavorable, 30% as neither unfavorable nor favorable, 10% as rather favorable, and 2% as definitely favorable. Therefore, the overall perception of pandemic for the childbearing appears as unfavorable.

The end of April (the time of the collection of the data) signified the end of the strictest COVID-19 restrictions. Previous individual experiences of the pandemic (the worsening or improvement of Job & Finances and Family life) may have affected the subjective evaluation of the favorability of the situation in April 2021 in terms of childbirth. Table 1 illustrates that the improvement in Job & Finances was associated with the higher odds (odds ratio (OR) 1.524 for men and OR 1.414 for women) of the considered pandemic period being seen as favorable for childbirth. Moreover, in the case of women, an improvement in Family life was associated with the higher odds (OR 1.480) of evaluating the considered period positively in terms of childbirth.

### Individual experiences of the pandemic and reproductive behavior

As indicated in the previous section, individual experiences of the pandemic influenced the evaluation of the favorability of the considered period for childbirth. The following analysis

**Table 1. Odds ratios for the evaluation of the pandemic in April 2021 as favorable for childbirth dependent on individual experiences of the pandemic (worsening/improvement of job & finances and family life).**

|  | MEN | | | WOMEN | | |
|---|---|---|---|---|---|---|
|  | exp(B) | 95% CI for exp(B) | sig. | exp(B) | 95% CI for exp(B) | sig. |
| **Constant** | 0.021 |  | 0.329 | 0.000 |  | 0.000 |
| Control variables: |  |  |  |  |  |  |
| **None or one child** (ref. 2+ children) | 1.173 | (0.601–2.291) | 0.559 | 0.640 | (0.732–2.758) | 0.299 |
| **Age** | 1.136 | (0.740–1.744) | 0.559 | 1.434 | (0.895–2.296) | 0.134 |
| **Age*age** | 0.998 | (0.992–1.004) | 0.519 | 0.995 | (0.988–1.001) | 0.111 |
| Independent variables: |  |  |  |  |  |  |
| **Job & finances** | 1.524 | (1.089–2.134) | 0.014 | 1.414 | (1.044–1.915) | 0.025 |
| **Family life** | 1.227 | (0.889–1.694) | 0.213 | 1.480 | (1.100–1.989) | 0.010 |
| Included N | 341 | | | 441 | | |
| Negelkerke R2 | 0.050 | | | 0.089 | | |

Data: Czech GGS COVID Pilot—Follow-up Study (Kreidl et al. 2021b), N = 782.

shows the extent to which the evaluation of favorability and individual experiences of the pandemic were associated with fertility behavior.

**Reduction in overall fertility intentions.** The literature indicates that the pandemic may have led to the postponement of childbirth or even the abandoning of reproductive plans. The association with the reduction of reproduction plans was analyzed depending on both perceived favorability and personal past experience. It has already been shown that the evaluation of favorability is, to some extent, associated with past experience of the pandemic and that the degree of association differs between men and women. The following model considers both these factors (three variables) together since the component of the favorability evaluation that is unrelated to experience of the pandemic can be influential in terms of reproductive behavior.

7% of both the male and female respondents declared that they had reduced the intended number of children due to the pandemic. Table 2 indicates that the odds ratio for men concerning a reduction in reproductive plans due to the pandemic was lower in terms of

**Table 2. Odds ratios for reducing reproductive plans depending on favorability and individual experiences of the pandemic.**

|  | MEN | | | WOMEN | | |
|---|---|---|---|---|---|---|
|  | exp(B) | 95% CI for exp(B) | sig. | exp(B) | 95% CI for exp(B) | sig. |
| **Constant** | 0.204 |  | 0.730 | 0.040 |  | 0.421 |
| Control variables: |  |  |  |  |  |  |
| **None or one child** (ref. 2+ children) | 3.225 | (0.962–10.814) | 0.058 | 1.345 | (0.595–3.041) | 0.477 |
| **Age** | 0.974 | (0.578–1.642) | 0.921 | 1.130 | (0.727–1.758) | 0.587 |
| **Age*age** | 1.000 | (0.993–1.007) | 0.983 | 0.998 | (0.992–1.004) | 0.494 |
| Independent variables: |  |  |  |  |  |  |
| **Job & finances** | 0.475 | (0.298–0.758) | 0.002 | 0.749 | (0.522–1.076) | 0.118 |
| **Family life** | 0.633 | (0.394–1.019) | 0.060 | 0.839 | (0.591–1.191) | 0.327 |
| **Favorability** | 0.657 | (0.380–1.135) | 0.132 | 0.600 | (0.380–0.947) | 0.028 |
| Included N | 341 | | | 441 | | |
| Negelkerke R2 | 0.220 | | | 0.073 | | |

Data: Czech GGS COVID Pilot—Follow-up Study (Kreidl et al. 2021b)

**Table 3. Odds ratios for current attempts to conceive depending on favorability and individual experiences of the pandemic.**

| | MEN | | | WOMEN | | |
|---|---|---|---|---|---|---|
| | exp(B) | 95% CI for exp(B) | sig. | exp(B) | 95% CI for exp(B) | sig. |
| Constant | 0.000 | | 0.006 | 0.000 | | 0.005 |
| Control variables: | | | | | | |
| None or one child (ref. 2+ children) | 5.299 | (1.926–14.58) | 0.001 | 3.603 | (1.373–9.453) | 0.009 |
| Age | 3.505 | 0.018 | 0.018 | 12.199 | (1.912–77.848) | 0.008 |
| Age*age | 0.983 | (0.969–0.997) | 0.018 | 0.965 | (0.940–0.991) | 0.008 |
| Independent variables: | | | | | | |
| Job & finances | 0.926 | (0.584–1.468) | 0.743 | 0.788 | (0.512–1.213) | 0.280 |
| Family life | 1.097 | (0.656–1.833) | 0.725 | 0.897 | (0.589–1.367) | 0.613 |
| Favorability | 1.327 | (0.843–2.090) | 0.222 | 1.107 | (0.713–1.720) | 0.651 |
| Included N | 263 | | | 372 | | |
| Negelkerke R2 | 0.180 | | | 0.192 | | |

Data: Czech GGS COVID Pilot—Follow-up Study (Kreidl et al. 2021b); 78 men and 69 women were excluded since they had no partner at the time of the collection of the data.

improvements in Job & Finances (OR 0.475). For women, the optimistic evaluation of the considered pandemic period for childbirth acted to lower the odds ratio (OR 0.600) of reducing their childbirth plans.

**Current attempts to conceive.** In a certain way, current attempts to conceive represents the reverse process to the postponement or abandonment of reproductive plans. 11% of the male and 7% of the female respondents declared that they had attempted to conceive with their partner during the period of data collection (that is at the end of the pandemic most restrictive period, in April 2021). As shown in Table 3, individual experiences of the pandemic (Job & Finances and Family life) and the subjective evaluation of the favorability of the pandemic for childbirth did not act to change the odds ratio for attempting to conceive for either the men or the women at the time of data collection (April 2021).

**Short-term fertility intentions.** Despite having no current intentions to conceive, neither the men nor the women respondents have necessarily abandoned their reproductive plans. One explanation is that they have merely postponed childbirth or intend to fulfil their fertility intentions within the next three years. 21% of the male and female respondents declared that they planned to have a child within the next three years.

The table (Table 4) presents the odds ratios for short-term fertility intentions (within the next three years). With respect to the final model, which takes into account experiences of the pandemic and childbirth favorability, only the optimistic evaluation of the favorability of the considered period for childbirth acted to change the odds ratios of having short-term fertility intentions. This finding was valid only for the male respondents, whose more optimistic evaluation of favorability is reflected in the higher odds of the realization of fertility intentions (OR 1.447).

## Discussion and conclusions

The study evaluated the effects of the COVID-19 pandemic on reproductive behavior in Czechia at the end of the period in which the strictest restrictions were applied and after the pandemic had lasted for more than one year [6]. The assumption of a negative impact was confirmed by the evaluation of the favorability of the considered period (April 2021) for having a

**Table 4. Odds ratios for short-term fertility intentions depending on favorability and individual experiences of the pandemic.**

|  | MEN | | | WOMEN | | |
|---|---|---|---|---|---|---|
|  | exp(B) | 95% CI for exp(B) | sig. | exp(B) | 95% CI for exp(B) | sig. |
| **Constant** | 0.000 |  | 0.001 | 0.000 |  | 0.000 |
| Control variables: |  |  |  |  |  |  |
| **None or one child** (ref. 2+ children) | 5.298 | (2.486–11.291) | 0.000 | 5.915 | (3.108–11.257) | 0.000 |
| **Age** | 1.928 | (1.184–3.138) | 0.008 | 2.875 | (1.754–4.715) | 0.000 |
| **Age*age** | 0.990 | (0.983–0.997) | 0.005 | 0.982 | (0.975–0.990) | 0.000 |
| Independent variables: |  |  |  |  |  |  |
| **Job & finances** | 0.998 | (0.731–1.362) | 0.991 | 0.876 | (0.662–1.160) | 0.355 |
| **Family life** | 1.325 | (0.962–1.824) | 0.085 | 0.809 | (0.613–1.066) | 0.132 |
| **Favorability** | 1.447 | (1.054–1.987) | 0.022 | 1.279 | (0.946–1.729) | 0.110 |
| Included N | 329 | | | 422 | | |
| Negelkerke R2 | 0.247 | | | 0.419 | | |

Data: Czech GGS COVID Pilot—Follow-up Study (Kreidl et al. 2021b); 12 men and 19 women were excluded from the analysis since they (or their partners) were pregnant at the time.

child based on data from the Czech GGS COVID Pilot—Follow-up study [9]. In general, the pandemic period was confirmed as being rather unfavorable for the birth of a child (60% of the female and 54% of the male respondents considered this period to be unfavorable). However, the results were not entirely uniform, and it appeared that a proportion of the population perceived the pandemic period as favorable for childbirth (12% of the female and 15% of the male respondents). It is likely that the perception of the pandemic period reflected past experience, current status and future expectations, thus reflecting the future narrative. While for men, improved financial and working conditions played a role in the positive evaluations of the pandemic period in terms of childbirth, for women, these factors were supplemented with the role of family relationships. This can be considered from the perspective that it is seen as important for men to materially secure the family, while women are generally more sensitive to the development of family relationships. This can be further explained by the assumption that the increasing need for care for the household and children (or other family members) affects women more than their partners [16], as confirmed by a number of other studies [17–20].

The association of previous experience of the pandemic and perceptions of the pandemic period was observed with respect to the reduction of overall reproductive plans. Previous studies that focused on the reasons for reductions in reproduction [21, 22] determined that the quality of the partner relationship, as well as job and income stability comprise important associated factors. This paper examined the issue from an alternative perspective that focused more on what acts to prevent reductions in reproduction. Seven percent of the respondents were found to have reduced their overall reproductive plans as a result of the pandemic. While the results suggest that for men the risk of a reduction in reproductive plans is lower following an improvement in economic stability, for women a lower risk of a reduction in reproductive plans was associated with the perception of the pandemic period as favorable for the birth of a child. It can be concluded, based on these results, that the men's decisions on the total planned number of children were rationalized (based primarily on their economic situation), while those of the women were based on considerably more complex assessments that took into account the material situation, family relationships and perceptions of the future. These findings are thus consistent with existing knowledge on the causes of reductions in reproductive

plans and are, moreover, consistent with the thesis of Voicu and Badoi [16], since favorability may also have been reflected in the fact that the pandemic period reduced other realization opportunities and, thus, motherhood was reflected in a reduction in the opportunity costs concerning career progression. Of course, this only applies to a select group of the female population that, for example, is able to work from home, which is consistent with the claim that the impact of the pandemic varied according to occupation [32].

Thus, although the overall reproductive plans of both men and women appear to have been related to the pandemic, current attempts to conceive are not. It seems that the decision to have a child is not related exclusively to a deterioration or improvement in the life situation due to the pandemic or perceptions thereof. A further possibility is that the pandemic acted to both improve and worsen the life situations of individuals, which closely reflects the results of studies that have addressed, for example, the effects of employment on women's reproductive plans. In some cases, a loss of employment is seen as an opportunity to finally realize one's reproductive plans, while in others it adds to economic instability, which leads to the postponement of fertility intentions [37, 38]. Alternatively, a stable job with good career prospects may provide a sense of financial security that increases the probability of realizing one's fertility plans; however, conversely, the resulting increase in the opportunity cost of parenthood may lead to the postponement of parenthood until the opportunity cost is lower.

Finally, the analysis of the intention to have a child in the next three years revealed that 21% of respondents intended to do so. For both men and women, the changes that occurred in terms of the material situation and family relations did not affect short-term fertility intentions. However, a positive association was confirmed for the male respondents, who considered the most restrictive pandemic period (April 2021) to be favorable for the birth of a child; hence, the chances of their planning to have a child within three years were enhanced. Thus, while the male respondents stated a preference for a reduction in the planned number of children based on their rational evaluation of the economic situation, the planning of a child in the short term was associated with a more complex evaluation of the current situation in terms of childbirth.

The covid-19 pandemic period significantly changed not only society as a whole but also the lives of individuals. The results presented revealed that the reproductive behavior of individuals may change during difficult times. Although the quality of interpersonal relationships affects the decision-making process, material insecurity also contributes significantly to the final decision. Thus, it is important that even in times of crisis, be it due to pandemics or adverse economic conditions, the availability of assistance measures for economically active individuals and families is not reduced.

In conclusion, it is important to note that the fact that our study does not track paired data represents a significant shortcoming. Since reproduction is the result of a joint decision by the man and the woman, the observation of how the life trajectories of the men and the women during the pandemic affected their reproductive behaviour would add significant extra value to the research. Nevertheless, in addition to current efforts at conception, which can be assumed to follow the agreement of both partners, the intentions of men and women concerning the planning of the total number of planned children and childbirth in the next three years may differ, thus justifying the conducting of studies based on individual, unpaired data.

## Author Contributions

**Formal analysis:** Jitka Slabá.

**Writing – original draft:** Jitka Slabá.

**Writing – review & editing:** Jitka Slabá.

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
