## [Decision Letter · Decision Letter 0]

20 Jun 2023

PONE-D-22-30353

Changes in reproductive behavior associated with the perception and individual experiences of the COVID-19 pandemic

PLOS ONE

Dear Dr. Slabá,

Thank you for submitting your manuscript to PLOS ONE. After careful consideration, we feel that it has merit but does not fully meet PLOS ONE’s publication criteria as it currently stands. Therefore, we invite you to submit a revised version of the manuscript that addresses the points raised during the review process.

While the two referees recommend acceptance they have suggestions for a minor revision and clarification. In addition, I would like clarification on why the second wave was not used to explain fertility decisions based on the predictive content of perceptions a few months earlier.

We look forward to receiving your revised manuscript.

Kind regards,

José Antonio Ortega, Ph.D.

Academic Editor

PLOS ONE

Journal Requirements:

This output was supported by the NPO "Systemic Risk Institute" number LX22NPO5101, funded by European Union - Next Generation EU (Ministry of Education, Youth and Sports, NPO: EXCELES) and the Charles University UK UNCE/HUM/018 programs. 

Reviewers' comments:

Reviewer's Responses to Questions

**Comments to the Author**

1. Is the manuscript technically sound, and do the data support the conclusions?

Reviewer #1: Yes

Reviewer #2: Yes

2. Has the statistical analysis been performed appropriately and rigorously? 

Reviewer #1: I Don't Know

Reviewer #2: I Don't Know

3. Have the authors made all data underlying the findings in their manuscript fully available?

Reviewer #1: Yes

Reviewer #2: Yes

4. Is the manuscript presented in an intelligible fashion and written in standard English?

Reviewer #1: Yes

Reviewer #2: Yes

5. Review Comments to the Author

Reviewer #1: It is a useful piece of work that shows the relationality between factors like job security, income opportunities, and gendered perceptions with the favourability for childbirth. The paper touches upon the disruption of certainty and the apprehensions about “what will happen now to our society?” in the pandemic years, and goes on to build a case about reproductive behaviours.

The lack of uniformity in the data findings, to me, represents a strength: that reproductive behaviour can not always be quantified and rationally explained, and the ruptures there have a story to tell about social, individual, past, and future relationships. That also exhibits how he life trajectories of the men and women differed and influenced their reproductive behaviour, wants about number of children, the attempts to conceive, and plans to have children within the next 3 years. For instance, it was interesting to see how men and women assessed changes to their financial and job security in relation to their reproductive behaviour.

The findings and the discussion did not quite address this particular assumption of the authors though: "that social distancing has a positive effect on the fertility intentions of women and suggest that the corresponding increased need for caring for the child and the household affect primarily women, thus pressurizing women into reverting to the traditional family model." A better explanation for this assumption would be rather crucial for the paper, I believe.

Reviewer #2: This study brings out interesting findings on gendered differences in reproductive behaviours. That while women viewed this period as unfavourable for reproduction, men viewed it as favourable.

In the introduction, the study mentions that the pandemic restrictions "allowed for a deepening of relationships between partners and increased involvement of both parents in caregiving". However, there were many reports on increased household and caregiving burdens on women, and increased gender based violence in homes during the pandemic across high income and low income countries. These important factors which would impact reproductive choices and behaviours, while touched upon in the introduction, are not discussed in detail in the study.

The study mentions that the pandemic period saw a reduction in opportunity costs concerning career progression for women depending on occupation. Is the data collected in the study segregated by occupation type. Would be beneficial to substantiate this with study data.

Are there any long-term repercussions of the findings presented in the study? Or implications for policy?

6. PLOS authors have the option to publish the peer review history of their article (what does this mean?). If published, this will include your full peer review and any attached files.

Reviewer #1: No

Reviewer #2: No

---

## [Author Response · Author response to Decision Letter 0]

26 Jun 2023

The responses are in the attached file. I copy it also here.

On your editorial comment

E: „In addition, I would like clarification on why the second wave was not used to explain fertility decisions based on the predictive content of perceptions a few months earlier.

Not sure that I understand well your question, so I am paraphrasing it in my own words in response. The pilot survey (what we might understand here as the first wave) ran from December 2020 to February 2021, a period that is inconsistent in terms of pandemic restrictions. This pilot sample contained questions that are part of the uniform Generation and Gender Survey, where the fertility section is relatively brief and very general, and thus the effect of the pandemic or individual life course is hard to distinguish. Whereas the Follow-up conducted in April (second wave) allowed questions to be asked that already reflected the pandemic itself and additionally asked about the declared reduction in reproductive plans due to the pandemic. Thus, both surveys (first and second waves) included only questions about short-term fertility intentions and current attempts to conceive. Initially, we even considered comparing the development of these intentions over time within the research team, but very problematic here is the very inconsistent spacing of the first and second wave (2 months vs 4 months) with the different impact of the pandemic, and we do not know whether those original intentions (first wave) have already been affected by the pandemic. 

Is this the clarification you asked for?

I very much appreciate the comments from both reviewers who revealed a weakness of the manuscript, where the theoretical part dropped the reflection on the division of labor in the household focusing on childcare, which was already reflected in the introduction and conclusion of the manuscript.

Specifically:

1)

R1: The findings and the discussion did not quite address this particular assumption of the authors though: "that social distancing has a positive effect on the fertility intentions of women and suggest that the corresponding increased need for caring for the child and the household affect primarily women, thus pressurizing women into reverting to the traditional family model." A better explanation for this assumption would be rather crucial for the paper, I believe.

R2: In the introduction, the study mentions that the pandemic restrictions "allowed for a deepening of relationships between partners and increased involvement of both parents in caregiving". However, there were many reports on increased household and caregiving burdens on women, and increased gender based violence in homes during the pandemic across high income and low income countries. These important factors which would impact reproductive choices and behaviours, while touched upon in the introduction, are not discussed in detail in the study.

I reflected on these two points by rewriting the relevant paragraph in the theoretical framing of the study.

Original version: However, the anti-pandemic measures limited these options to a significant extent, and it is not unreasonable to assume that for part of the population, starting (or expanding) a family moved up many people’s list of priorities. This assumption is supported by a study by Voicu and Badoi [13], which examines the possible effects of the pandemic on fertility from the viewpoint of gender roles. The authors assume that social distancing has a positive effect on the fertility intentions of women and suggest that the corresponding increased need for caring for the child and the household affect primarily women, thus pressurizing women into reverting to the traditional family model. On the other hand, limited opportunities for self-realization as well as changes in the life situations of individuals in terms both of the quality of the partnership [14,15] and employment and income stability [4] may result in not only in a preference for parenthood but also in the case of the deterioration may have acted to reduce or delay reproduction.

New version: However, the anti-pandemic measures limited these options to a significant extent, and it is not unreasonable to assume that for part of the population, starting (or expanding) a family moved up many people’s list of priorities. In addition to the change in priorities [13], the enhancement of the reconciliation of work and family responsibilities may have contributed positively to fertility attitudes, for example via the option to work from home [13,14] and the forced reduction of working hours, which comprised an official supportive measure in many European countries during the pandemic [13,15]. Voicu and Badoi [16] suggest that this reconciliation may have increased the fertility of women who were forced to stay at home to care for their children due to the pandemic, and thus were forced to take on the position of caregiver due to the reduced availability of formal and informal childcare services [5]. It should be noted here that although the extra demands for childcare impacted both mothers and fathers, most of the extra responsibilities were shouldered by women [17,18,19,20]. On the other hand, the extra burden of caring for children, together with the limited opportunities for self-realization, as well as changes in the life situations of individuals in terms both of the quality of the relationship [21,22] and employment and income stability [4] may have resulted not only in a preference for parenthood but also, in the case of the deterioration of conditions, may have acted to reduce or delay reproduction.

I hope that now the consideration of the impact of household arrangements on fertility intentions is clear. 

I have also paid close attention to the comment on domestic violence by the second reviewer; in studying the available literature, I am unable to determine for certain whether domestic violence affects partners' reproductive intentions. I am aware that there has been an increase in domestic violence during the pandemic (in the case of the Czech Republic through NGOs providing support), I believe this is part of the assessment of relationship quality, which is mentioned by the manuscript. 

2)

R2: The study mentions that the pandemic period saw a reduction in opportunity costs concerning career progression for women depending on occupation. Is the data collected in the study segregated by occupation type. Would be beneficial to substantiate this with study data.

I completely understand this idea and it was one of my initial considerations as the GGS study has ISCO codes for individual respondents. In the final version of the manuscript, I moved away from assessing these individual characteristics (such as education or job position), partly because this would have reduced the sample so much that the test results would no longer be reliable, and secondly, it would still be burdened with considerable error, since the very determination of which occupations were advantaged or disadvantaged by the pandemic would be very difficult.

3)

R2: Are there any long-term repercussions of the findings presented in the study? Or implications for policy?

I would like to thank the reviewer for reminding me this. The following paragraph has been added to the Conclusion.

New paragraphs: The covid-19 pandemic period significantly changed not only society as a whole but also the lives of individuals. The results presented revealed that the reproductive behavior of individuals may change during difficult times. Although the quality of interpersonal relationships affects the decision-making process, material insecurity also contributes significantly to the final decision. Thus, it is important that even in times of crisis, be it due to pandemics or adverse economic conditions, the availability of assistance measures for economically active individuals and families is not reduced.

---

## [Editor Report · Decision Letter 1]

6 Jul 2023

Changes in reproductive behavior associated with the perception and individual experiences of the COVID-19 pandemic

PONE-D-22-30353R1

Dear Dr. Slabá,

We’re pleased to inform you that your manuscript has been judged scientifically suitable for publication and will be formally accepted for publication once it meets all outstanding technical requirements.

Kind regards,

José Antonio Ortega, Ph.D.

Academic Editor

PLOS ONE

Additional Editor Comments (optional):

The proposed changes strengthen the paper and it is not needed to send the manuscript back to the reviewers, and yes, that was the clarification I needed. I believe the paper is ready for publication. 
---

## [Editor Report · Acceptance letter]

10 Jul 2023

PONE-D-22-30353R1 

Changes in reproductive behavior associated with the perception and individual experiences of the COVID-19 pandemic 

Dear Dr. Slabá:

I'm pleased to inform you that your manuscript has been deemed suitable for publication in PLOS ONE. Congratulations! Your manuscript is now with our production department. 

Kind regards, 

on behalf of

Dr. José Antonio Ortega 

Academic Editor

PLOS ONE